# Development and Optimization of 3D-Printed Flexible Electronic Coatings: A New Generation of Smart Heating Fabrics for Automobile Applications

**DOI:** 10.3390/mi14040762

**Published:** 2023-03-29

**Authors:** Léopold Diatezo, Minh-Quyen Le, Christine Tonellato, Lluis Puig, Jean-Fabien Capsal, Pierre-Jean Cottinet

**Affiliations:** 1Electrical Department, Ladoua Campus, University Lyon, INSA-Lyon, LGEF, EA682, F-69621 Villeurbanne, France; 2Company TESCA-Group, 17452 Massanes, Spain

**Keywords:** printed electronic coatings, smart heating textiles, numerical simulation, optimization design, thermal and electrical characterizations, automobile applications

## Abstract

Textile-based Joule heaters in combination with multifunctional materials, fabrication tactics, and optimized designs have changed the paradigm of futuristic intelligent clothing systems, particularly in the automobile field. In the design of heating systems integrated into a car seat, conductive coatings via 3D printing are expected to have further benefits over conventional rigid electrical elements such as a tailored shape and increased comfort, feasibility, stretchability, and compactness. In this regard, we report on a novel heating technique for car seat fabrics based on the use of smart conductive coatings. For easier processes and integration, an extrusion 3D printer is employed to achieve multilayered thin films coated on the surface of the fabric substrate. The developed heater device consists of two principal copper electrodes (so-called power buses) and three identical heating resistors made of carbon composites. Connections between the copper power bus and the carbon resistors are made by means of sub-divide the electrodes, which is critical for electrical–thermal coupling. Finite element models (FEM) are developed to predict the heating behavior of the tested substrates under different designs. It is pointed out that the most optimized design solves important drawbacks of the initial design in terms of temperature regularity and overheating. Full characterizations of the electrical and thermal properties, together with morphological analyses via SEM images, are conducted on different coated samples, making it possible to identify the relevant physical parameters of the materials as well as confirm the printing quality. It is discovered through a combination of FEM and experimental evaluations that the printed coating patterns have a crucial impact on the energy conversion and heating performance. Our first prototype, thanks to many design optimizations, entirely meets the specifications required by the automobile industry. Accordingly, multifunctional materials together with printing technology could offer an efficient heating method for the smart textile industry with significantly improved comfort for both the designer and user.

## 1. Introduction

Creating smart textiles is not a new goal; however, understanding and optimizing their characteristics still requires deep investigations for successful scale up to the industrial scale. For more than twenty years, multifunctional materials, conductive composites, ferroelectric polymers, and shape memory metal alloys have been studied for this issue [1,2,3,4,5,6,7,8,9,10]. To differentiate themselves and conquer the markets, manufacturers are seeking to develop products with further added value. For instance, Zhu et al. and Li et al. developed high-performance flexible transparent electrodes [11,12] and transparent glass heaters [13,14] with the embedded metal mesh fabrication technique using electric-field-driven microscale 3D printing. Regarding applications in artificial intelligence, electronic skin, and human health monitoring based on flexible wearable electronics, Zhang et al. developed a novel surface reconstruction strategy towards breathable, flexible, highly conductive, bark-shaped MXene/textiles [15,16]. The progress of materials defined as multifunctional is probably the major driver behind this evolution, where functions are explored and integrated to make the products intelligent [17,18,19,20,21,22,23]. Following the example of mechatronics combined with mechanical, computational, and electronic functions, smart fabrics have found primary applications in recent years by integration with multifunctional materials [24,25,26,27,28,29]. In the context of this research, we particularly focus on the heating applications of these materials, with the aim of developing a new generation of smart car seat fabrics for automobile applications.

Among multifunctional materials, polymer matrices, with their light weight and mechanical flexibility, are promising candidates for heat dissipation systems [30,31]. However, polymeric materials have an intrinsically low thermal conductivity [32,33], which drastically restricts their practical use in heating car seat fabrics. To improve their thermal conductivity and make them more suitable for real-world situations, various studies have frequently incorporated additives with high thermal conductivities, such as silver (Ag) [34], copper (Cu) [35], carbon nanotubes [36], aluminum (Al) [37,38], graphite (GR) [39], and graphene oxide [40]. The formation of thermal conductive pathways facilitates heat diffusion throughout the composites, which is the key to achieving enhanced thermal transfer in polymer-based composites [38,41].

In addition, metal and carbon fillers have an excellent ability to conduct electricity; thus, a small concentration of these fillers in polymers may result in a high electrical conductivity of the composites [42,43,44,45]. An appropriate choice of polymer matrix and incorporated conductive particles can lead to the creation of composite coatings with enhanced thermal and electrical properties [46,47,48]. When two or more constituents are combined to form a layered or mixed structure, the features of a traditional polymer-based coating can be refined to address specific requirements. The developed composite coatings aim to convert the fabric substrates into electrically and thermally conductive materials, without greatly altering the existing substrate properties. Different methods comprising electroless plating, evaporative deposition, and sputtering have been frequently used to coat the outside of fabrics [49]. Recently, an alternative method to develop heating fabrics is presented by 3D printing additive manufacturing (AM), where metal fillers (e.g., copper, silver, gold, carbon, etc.) can be injected into conventional polymer-based inks [50,51]. Moreover, the AM approach is a simple way to obtain solid patterns of conductive inks on textile substrates, i.e., usually porous when produced with knitted or woven yarn. 

Therefore, 3D-printed coatings, which are expected to improve the conventional fabrication approach of heated fabrics, are investigated in this research. Actually, standard car seat heating technology relies on resistive elements carrying electrical current to generate heat. Though this process is quite effective, it requires bulky power cables that must be covered in insulating foam to stop the seat from becoming unpleasantly hot, and it can take an impractically long time for seats to heat up. There is also a certain risk of overheating, and thus a thermostat is needed to keep the temperature within the correct range. Printed electronic coatings promise to overcome these challenges and are starting to be used to heat car interiors. Since the printed elements only heat the surfaces that are in close proximity to the body, the energy consumption is significantly reduced. In addition, compared to traditional seat heating systems, printed coatings are thinner and lighter, meaning less padding and foam is required, leading to a considerably faster heat-up time. Furthermore, these heating elements can be printed on flexible materials such as textiles, making them easy to incorporate into car designs, and thus improving sustainability.

It is obvious that textile-based printed electronics open new opportunities for next-generation smart heating devices. Nonetheless, an essential understanding of the characteristics of material coatings is still needed to accelerate their large-scale production. Therefore, this work thoroughly investigates the design optimization of the printed composite coatings, together with the characterizations of their morphological, electrical, and thermal properties. To date, such investigations have rarely been reported in the literature to the best of our knowledge. The aim of this project is to develop the first printed heater prototype that matches the specifications imposed by the automobile industry (Tesca group). Numerical calculations together with empirical measurement are carried out, confirming the reliability of the proposed approach. Thanks to the significant optimization of the conductive patterns, it is possible to achieve a system with excellent performance, including a high heating efficiency, good temperature homogeneity, and fast time response, while avoiding overheating. 

## 2. Material Process and Design Architecture

### 2.1. Materials Selection

Conductive ink is the most important component in the printing of metallic structures. Several conductive materials could be considered for this purpose, such as conductive polymers [52], carbon [53,54], organic/metallic compounds [53], metal precursors [55], and metal nano/micro particles [56]. Conductive polymers are low cost and can be used on large surfaces that need high flexibility, but they also have a lower performance because of their weak electrical conductivity compared to their pure metal precursors [57,58]. On the other hand, the use of metal precursors requires an additional heat treatment (>250 °C) to reduce the precursors to metallic species, which is not appropriate for printing on flexible structures. Organic/metallic compounds seem to be an alternative solution that achieves the best compromise between mechanical and electrical performance. Most commercially available conductive inks are thus based on compounds of silver, gold, carbon particles, or of other metals such as copper, which is cheap and easily dispersed in water. Gold ink is quite expensive so is not further considered in this study. 

Accordingly, three commercially available conductive inks were selected: silver solvent-based ink (namely an Ag/BGA composite) and copper and carbon water-based inks (namely Cu/PU and C/PU composites). Recently, these conductive materials have been extensively used for applications related to printed electronics and functional 3D printing. Moreover, they are adaptable to various types of flexible substrates such as polyamide, polyethylene terephthalate (called PET or polyester), and photographic paper, among others. Table 1 summarizes the fundamental properties of the three chosen inks that can be found in the manufacturer’s technical documents. As it can be seen, the Ag/BGA composite exhibits an extremely low viscosity, suggesting that it would penetrate a textile substrate more easily than the other two inks. Consequently, the continuity of the printed track could be interrupted, particularly with a very thin silver coating. In fact, the viscosity has been revealed to be a critical parameter that can have a strong impact on the printing shape fidelity during the AM procedure. As mentioned in our previous works [59,60], the ideal range of the viscosity is around 0.5–50 Pa.s for screen printing process, and similar values was found for the 3D extrusion-based printing technique [61]. Accordingly, the Ag/BGA coating, with its particularly low viscosity (i.e., out of the ideal range), should be printed in multilayers to avoid discontinuity of the electrical contact. Eventually, to efficiently render the printed track conductive, a thermal annealing posttreatment is performed for all inks. This step might limit the type of substrates that can be used, or otherwise impair the conductivity of the printed line. 

The three composite inks were then printed on flexible fabrics with foam linings provided by the Tesca group. The optimal coated fabric was next integrated into a tablecloth made of polyester. Actually, more than 90% of all car seat fabric in the world is covered by polyester, because only this fiber combines the required standards of high toughness, reasonable cost, low mass density, high abrasion, and UV resistance [62].

### 2.2. Printing Process of the Conductive Coating

The printing setup consists of a modular head SDS extruder (cold flow at room temperature) and a glass platform. The syringe pump extruder has a standard ink reservoir of 30 mL that can print fluids in desired patterns and shapes using an appropriate nozzle diameter of a stainless-steel nozzle tip, the inner diameter of which equals 0.4 mm. Repetrel plug-in controlling software was used to run most of the Hyrel equipment, enabling the generation of G-code from various 3D designs (e.g., .stl or .obj file, text file, etc.) and other system control functions. For instance, the desired shapes and sizes of the conductive tracks could be built by the Computer Assistant Design (CAD) software. The models were then fed into Slic3r which converts the 3D models into G-code. To perform the desired printing trajectories, relevant parameters affecting the printing quality, such as layer height, ink flowrate, line density, pressure control, etc., were carefully tuned in the Slic3r software. After several adjustments, the best flow rate and pressure control were equal to 300 pulses/µL and 0.8 (1 is the default), respectively. Some other input parameters of the 3D printing system are presented in Table 2 [9]. 

For a better dispersion of the particles within the polymer solution, stirring was performed for 20 min. The mixtures were scattered by ultrasound treatment (Hielscher Ultrasonic Processor UP400S) with powerful impulsions for a few minutes until a perfectly homogeneous solution was obtained. Three conductive inks were then extruded, one at a time, onto the three separated fabric substrates that were lying on the printed bed. Once the printing process was complete, the sample together with its textile substrate were placed into an oven (Memmert V0 400 drying oven) at 170 °C for 2 min to initiate cross-linking polymerization (on recommendation from the ink’s suppliers). To simplify the characterization tests (Section 3), all printed coatings were made in a rectangular shape of 150 mm length and 16 mm width. Figure 1 shows the images of four conductive lines coated on the foam lining fabrics, which stack compactly after the curing processes. Each conductive line consists of one or several printed thin layers, i.e., an approximately 200 µm/layer. To avoid any oxidation of the copper electrode, a PU polymer of around 100 µm thick was deposited on the copper coating. For the carbon coating, add-on protection was not necessary as no oxidation occurred.

Figure 2 illustrates the whole 3D printing setup. After printing, the coating inks were thermally treated in an oven (Votsch Industrietechnik TM, VT7004, Weiss Technik GmbH, Reiskirchen-Lindenstruth, Germany) at a temperature of 80 °C for 5 min. To stop the oxidation reaction of the copper wire electrode, a thin layer of PU (~100 µm thick) was deposited on the copper surface. The quality of the conductive inks together with the accuracy of the printing process have a strong impact on the electrical and thermal performances [64]. 

## 3. Optimization of Design Architecture

### 3.1. Specifications and Design Rule

The heating fabric was designed to be integrated inti a tablecloth (i.e., assimilated to a car seat), which must fit the following specifications defined by Tesca:Generated temperatures between 40 °C and 43.5 °C on the back side of the heating surface;Good homogeneity between different heating areas, with a maximum discrepancy of 0.5 °C;The response time in a transient regime (i.e., time needed to reach the target temperature) must not exceed 2 min;The occurrence of hot spots must be avoided at all costs (peak<45 °C).

In addition, the printed coatings (copper and carbon) should be flexible and as thin as possible so as not to alter the properties of the fabric substrate. As illustrated in Figure 3a, the 3D system design is composed of a rectangular fabric substrate onto which is printed a flexible electric heating circuit. The circuit consists of two principal copper electrodes directly connected to a DC supply voltage of 12 V and three heating resistors made of carbon composites. Both copper power buses are divided into three segments containing several sub-electrodes with identical sections, allowing to supply the same current intensity to all three resistors through the connection points. 

An example of the heater’s dimensions is shown in Figure 3b, where modification may be carried out to meet the design requirements. Only the dimensions of the copper principal electrodes (power bus) and the carbon patches are shown. All design information regarding the copper sub-electrodes (size, shape, number of subdivisions on each area, diameter of connection points, etc.) is not disclosed for confidentiality purposes. The geometrical features of the carbon patches were adapted to obtain a reasonable resistance at room temperature (around 5 to 10 Ω for each heating area) and to minimize the energy consumption of the tablecloth [65]. In order to enhance the temperature distribution, the three carbon elements have the same dimensions with a maximized heating surface, which is expected to achieve a homogeneous temperature for all areas. The width of the copper power bus should be large enough (~40 mm) to facilitate the current flow, which is in turn impeded by the thin thickness of the conductive coating (a few hundred µm). The resistances of the principal electrodes were designed to be equal to approximately 2 to 3 Ω, which is small enough to minimize the Joule losses. The subdivided electrodes, on the other hand, were built with a narrow width as they receive a much lower current intensity. As a result of their short length, the resistances of the sub-electrodes were assumed to be negligible, leading to a simplification in the equivalent electrical model of the fabric (Figure 3c). As seen, the copper power bus plays a key role in driving the electric current, while the carbon patches contribute to the electric–thermal conversion, with the aim of efficiently heating the tablecloth up to the target temperature.

The above analyses allow to achieve an initial design of the heating system with good compromise of the dimensions of both the copper power bus and the carbon resistors (Figure 4a). As later demonstrated in Section 3.2.2, two fundamental issues still persist in this design: the presence of significant lateral hot spots (overheating) and inhomogeneity of temperature among the three carbon areas. Since area 3 is closer to the power source, it is expected to receive a higher current intensity, thus resulting in a higher temperature than the other areas. Connections between the two materials of different resistances (copper and carbon, as displayed in Figure 4b) lead to the creation of hot spots, which are even more contrasting in the case of a high resistance discrepancy. This is due to the charge accumulation when flowing through a low-resistance conductor (copper) to a higher one (carbon). To solve these problems, we proposed two innovative designs focusing on the architecture enhancement of the subdivided electrodes, i.e., considered as key elements for the conversion between electrical power and thermal heating. 

The first design has an increased number of subdivided electrodes (denoted as *N*), with the intention of limiting the presence of hot spots that would damage the circuit and textile substrate in the long run. The current traveling in each sub-electrode (i.e., inversely proportional to *N*) is considerably reduced, and so is the charge accumulation at the contact points between the copper and the carbon coatings. To achieve the same current density in each area (to induce the desired temperature), the surfaces of all sub-electrodes are made equal to the one in the initial design (see Figure 4b,c).

Besides the matter of hot spots, solved by design 1, another solution dedicated to solving the heating homogeneity is considered in the second design. The goal of this design is to enhance the exchange surface of the subdivided electrodes (particularly for areas 1 and 2), while keeping the benefits of design 1 (Figure 4c). Therefore, some modifications of design 1 were made by increasing the size of the contact points as well as the number of sub-electrodes (*N*). Actually, it is not necessary for *N* to be identical to the three carbon areas to counterbalance the dissimilar distance between these areas and the electric source (as the Joule losses depend on the conductor length). 

In the following subsection, the three above designs (Figure 4) will be implemented in COMSOL Multiphysics software (version 5.2, COMSOL Multiphysics, Inc., Burlington, MA, USA). Their heating performances will then be analyzed and compared, allowing to verify whether or not they are fit for the intended purpose.

### 3.2. Design Optimization Based on FEM

This study focuses on optimization of the heating transfer of a tablecloth, which relies on numerical solutions of a finite element model (FEM). Usually faster and less expensive than physical experimentation, this method is powerful and of great interest in many application fields. In fact, the results of the numerical simulations allow choosing the optimal operating parameters while respecting the imposed specifications.

#### 3.2.1. Simulation Model Built in COMSOL

The FEM used in this study involves both electrical and thermal behaviors, which are suitably coupled to obtain a predictive 3D model of a tablecloth. The main geometrical and physical characteristics of the three materials used in the model are summarized in Table 3. The parameters of the conductive inks were deduced from experimental characterizations and analytical mixing rules, which will be detailed in Section 4 and Section 5. Some properties of the fabric substrate were found in the COMSOL library. 

The thermal radiation effect of the materials is considered: the emissivity (*ε*) is equivalent to 0.9 for the carbon ink and the substrate, while it is only 0.5 for the copper electrode. In order to evaluate the effect of dissipation induced by free natural air convection, the heat transfer coefficient ($h_{c}$) was kept constant at 5 W.m^–2^.K^–1^ [66,67]. The theory of thermal conduction and convection models used in the FEM have been previously developed in the literature [68,69,70]. 

The system was powered by a DC voltage source of 12 V connected to the copper electrodes, one of which is fixed at 12 V and the other is grounded. As shown in Figure 5, the mesh built for the tablecloth model was triangular in shape and heterogeneous, with a finer mesh at the electrode–tissue interface where the highest electrical gradients were expected. To obtain a high discretization quality of the modeled system, the mesh should be as fine as possible [71,72]. The problems encountered when refining the mesh were mainly related to the connection of the structure, but also to the limits imposed by the software or by the computer. As a result, a compromise between the accuracy of the numerical resolution and the calculation time must be considered.

#### 3.2.2. Optimization of the Heating Behavior in a Steady Regime

In a steady regime, the design was optimized in such a way that the temperature was homogenous for all three carbon areas (i.e., discrepancy of ≤0.5 °C), and the opposite surface of the tablecloth can be heated up to 43 °C. Another objective was to minimize the number of hot spots between the carbon patches and the copper electrodes so as to prevent burning. 

The first idea here is to modify one of the main physical parameters of the composite inks, e.g., the thermal conductivity (λ). Simulation tests were performed on the same initial design (Figure 4a) but with two configurations of $\lambda_{carbon}$ and $\lambda_{copper}$; one uses the real values obtained from the experimental characterizations of the materials (Figure 6a), while the other uses modified values, with the purpose of reducing the hot spots (Figure 6b). Regarding the color scale, Figure 6a clearly indicates a large disparity in the temperatures for the three areas, in which area 3 has the maximum temperature. Additionally, in this design, the appearance of noticeable hot spots on the lateral carbon patches are observed. Particularly, area 3 gives rise to an important peak of 48 °C, i.e., it greatly exceeds the allowed temperature limit. Increasing and equalizing the thermal conductivity of the copper and carbon inks $\left(\lambda_{carbon}=\lambda_{copper}=10\;W{.m}^{-1}.K^{-1}\right)$ can solve the hot spot problem, as suggested in Figure 6b. However, the homogeneity among the three areas is still not perfect (with 2 °C to 5 °C discrepancy), showing an inhomogeneous heatmap of the whole tablecloth surface. Furthermore, the temperature of all areas drastically drops, which is logical, as increasing the thermal conductivity of the materials leads to a lower temperature gradient for a given heat source rate. 

The above example has revealed that optimization based on changes in the coatings’ parameters could be promising, but several problems might be encountered in practice. The fact is that changing one parameter of the material could somehow influence other properties and generally compromises must be considered. For instance, increasing the thermal conductivity of the coatings requires composites with a higher particle concentration, which would dramatically affect their mechanical flexibility and fluid viscosity. This could be critical in certain approaches, especially in AM-based 3D printing. Besides, composite inks with the desired characteristics are not always commercially available. Customizing materials to obtain a new adequate formulation of ink is thus necessary. This matter is, however, costly and time consuming and needs thorough investigation in future work. 

Another solution relies on optimization of the coating design, as previously described in Section 3.1. In reality, more than 300 patterns were investigated to determine the most appropriate candidate fulfilling the requirements. For the sake of simplicity, only three representative designs are shown and compared, as illustrated in Figure 7. Here, D0, D1, and D2 correspond to the initial design (Figure 6b and Figure 7a), design 1 (Figure 7b), and design 2 (Figure 7c), respectively. As opposed to D0, D1 allows for a significant reduction in the number of hot spots, but the temperatures of the three areas are still not homogeneous. Remarkably, D2 (the most optimized design) definitively solves these two problems, and thus will be used for the development of a real prototype (cf. Section 4.4 and Section 5.4).

For an easier and more precise comparison, the numerical values are displayed in Table 4. The mean value $\left(T_{mean}\right)$ and the standard deviation (SD) of the temperature are computed from 40 data points acquired on each area surface. The relative variation (RV) shown in Table 4 is then deduced as:(1)$$RV=\frac{SD}{T_{mean}}\times 100\%$$

The relative deviation from the temperature of area 3 (denoted as *DT*3 and given by Equation (2)) is another indicator reflecting the temperature homogeneity of the three surfaces.
(2)$$DT3=\frac{T_{3}-T_{i}}{T_{3}}\times 100\%$$
where $T_{i}$ denotes the temperature of area i (with i=1,2,or 3).

Based on the results of Table 4, some remarks are outlined as follows:
Area 3 has the highest average temperature and dispersion of data, regardless of which design is selected.$T_{mean}$ increases from area 1 to area 3, explaining why *DT*3 of area 1 is at a maximum with respect to the other areas.A smaller value of *RV* compared to that of D0 leads to an improvement in the heat regularity of each area of D1, explaining the reduction in the hot spots. However, no change in *DT*3 indicates that, similar to D0, the heat transfer in D1 is not homogenous for all three areas. For all areas, D2 results in the smallest *DT*3 and *RV* compared to the other designs. This demonstrates that only D2 reaches expectations, as it is capable of dissipating heat evenly over all areas (maximum 0.6 °C discrepancy) and has small dispersion of data collected on each surface (~2%).

Figure 8a,b takes a closer look at the lateral and center temperature profiles, respectively, along the x-axis (length of the tablecloth) for all the three designs. It is observed that:The center profile seems to be more stable and homogenous than the lateral profile, in which hot spots appear caused by the connection of the subdivided copper electrodes.D1 and D0 have significant hot spots in area 3 in excess of 45 °C (i.e., beyond the allowed limit).In area 3, D1 and D0 have higher lateral and center temperatures than D2. This behavior is contrary to the other areas.Both the lateral and center profiles of D1 and D0 exhibit an obvious decrease along the length of the tablecloth, while those of D2 are almost constant.Only D2 succeeds in homogenizing the temperature of the three heating surfaces, as well as in preventing the presence of significant hot spots.

### 3.3. A First-Order Response in Transient Regime

In order to estimate the time needed to reach the target temperature, a study based on a transient regime was carried out. The time evolution of the heating temperature profile $\left(T\right)$ of the tablecloth (including fabric coated with three layers of carbon patches and two layers of copper electrodes) can be estimated by the following first-order heat transfer problem [73,74]:(3)$$\rho_{tab}VC_{p\_tab}\frac{\partial T}{\partial t}+\frac{1}{R_{th\_tab}}(T-T_{0})+h_{c}S(T-T_{0})=\Phi_{J}$$

The system is modeled as a linear RC thermal circuit in which $R_{th\_tab}$ and $C_{th\_tab}$ are, respectively, the thermal resistance and the specific capacity of the tablecloth; $\rho_{tab}$ denotes its mass density; *S* is the exchange surface between the structure and the surrounding air (corresponding to the three carbon patches); *V* denotes the heated volume (V=S×e, where *e* is the thickness of the tablecloth); $h_{c}$ denotes the heat transfer coefficient by natural convection; $T_{0}$ is the ambient temperature; and ${Փ}_{J}$ denotes the injected electrical power. Note that $R_{th,tab}$ is considered as the sum of the three thermal resistances corresponding to a multilayered (fabric/copper/carbon) structure. Based on the geometric parameters as well as the thermal conductivities of these three materials (Table 3), $R_{th\_tab}$ was estimated to be equal to 1.34 $W^{-1}.K$. As the volume of the fabric is much higher than that of the coating inks, while their mass densities are comparable, the weight fraction of the fabric is thus revealed to be the most important. Consequently, $\rho_{tab}$ and $C_{th\_tab}$ are close to the mass density and the specific capacity of the fabric, respectively. In other words, the copper and carbon coatings, because of their negligeable mass, do not significantly alter $\rho_{}$ and $C_{th}$ of the fabric substrate.

Regarding the model of Equation (3), the input power $\left({Փ}_{J}\right)$ is assumed to be composed of three terms. The first term (from left to right) is time dependent, and expresses the heat required to attain the temperature limit ($T_{l})$. When $T\to T_{l}$, there is no more variation in temperature and this term tends to 0. The second term corresponds to the heat flowing within and through the materials themselves from a hot to a cold surface. The last term refers to the free or natural convection heat transfer due to air motion in the presence of a temperature gradient $\left(T-T_{0}\right).$

Regarding the initial and the limit conditions, where $T\left(t=0\right)=T_{0}$ and $T\left(t\to \infty \right)=T_{l}$, the temperature in the transient regime follows the form of an exponential function, given by:(4)$$T(t)=(T_{0}-T_{l})e^{-t/\tau }+T_{l}$$
where $T_{l}$ corresponds to the temperature limit (in a steady state), which depends on ${Փ}_{J}$ and other material parameters, and *τ* denotes the time constant, expressed as:(5)$$\tau =\frac{\rho_{tab}VC_{p\_tab}}{h_{c}S+1/R_{th\_tab}}$$

For a comparison between the experimental and theoretical temperatures, data were collected from the carbon patch (area 2) using the optimized design (D2). As observed in Figure 9, the time response of both temperatures during the transient state exhibits a typical first-order trend, which agrees with the model in Equation (4) and with the results reported by Xiang et al. [75]. Empirical measurements show that the system reaches 95% of the temperature limit after around 112 s (the so-called response time), corresponding to three times the amount of the time constant (*τ*). This finding perfectly meets the requirements imposed by the Tesca group, where the time needed to attain the target temperature must not be in excess of 2 min. Reducing the material parameters, such as the thermal resistance and/or the specific heat, leads to a smaller value of *τ*, and thus a faster temperature response. Substituting the parameters in Table 3 into Equation (5) yields an estimated theoretical value of *τ*~330 s, i.e., three times greater than the real value, reflecting a discrepancy between the experiment and the numerical model. It was thus pointed out that the FEM is somewhat reliable in the permanent regime, where a good prediction of the static temperature is given. Inversely, the estimation of the time response in the transient state does not match with the real system to a certain extent.

In the following, a characterization method is performed to justify the parameters set in the FEM-based COMSOL software. Despite that the simulation and experimental tests only consider copper and carbon composite inks, the characterization tests additionally provide the properties of the silver ink. 

## 4. Characterization Methods

Thermal and electrical characterizations of the composite ink were conducted through experimental measurement that can be fitted with a theoretical model based on mixing rules. Table 5 presents the different properties of the polymer matrices (PU and BGA) and the conductive particles (Cu, Ag, and C), such as the thermal conductivity (λ), the specific heat ($C_{p}$), and the electrical conductivity (σ). These parameters are assumed to be constant within an operating temperature range of 0–50 °C. The mass density (ρ) of the polymer matrix and the particles, shown in the last column of Table 5, were used to determine the mass density ($\rho_{comp}$) and particle volume fraction (*v*) of the ink (indicated in the last two lines of Table 1), following the mixing rules:(6)$$\frac{1}{\rho_{comp}}=\frac{1-w}{\rho_{m}}+\frac{w}{\rho_{p}}$$
(7)$$v=\frac{\rho_{comp}-\rho_{m}}{\rho_{p}-\rho_{m}}$$
where *w* is the particle weight fraction of the different conductive inks, the values of which are given in Table 1. The subscripts *comp*, *m*, and *p*, respectively, denote the composite, matrix, and particles.

According to the literature [76], the electrical conductivities (*σ*) of the polymer matrices (e.g., PU and BGA) are within the range of 10^−12^–10^−8^ $S{.m}^{-1}$. This value, extremely small compared to that of the conductive particles, should not significantly influence *σ* of the composite. It is thus reasonable to take an average value of *σ* (~10^−10^
$S{.m}^{-1}$) for both PU and BGA matrices. 

**Table 5 micromachines-14-00762-t005:** General properties of matrices and particles used in the composite inks.

Materials	*λ* $\left(W{.m}^{-1}.K^{-1}\right)$	*C_p_* $\left(J{.kg}^{-1}.K^{-1}\right)$	*σ* $\left(S{.m}^{-1}\right)$	*ρ* $\left({g.cm}^{-3}\right)$
**Polymer matrix**	[77,78]	[79,80]	[76]	-
Polyurethane (PU)	0.23	1500	~10–10	1.13
Butyl Glycol Acetate (BGA)	0.19	2000	~10–10	0.94
**Conductive particles**	[81,82]	[83,84]	[85,86,87]	-
Copper particles (C)	138	720	~1.0 × 105	2.26
Carbon particles (Cu)	380	380	5.98 × 107	8.96
Silver particles (Ag)	420	240	6.30 × 107	10.4

### 4.1. Morphological Characterization

To visualize the surface state and the edge effect of the three conductive coatings, a morphological characterization via scanning electron microscopy (SEM) was performed. Observation tests were conducted on the top surface and the cross-section of the samples using compact SEM equipment (FlexSEM 1000II, Hitachi High-tech, Tokyo, Japan). These observations clearly allow for a better comprehension regarding the interaction between the ink/fabric interfaces, allowing for a justification of the material choice that is essential in the developed design. 

### 4.2. Thermal Characterization

A specific device (Jeulin, Ref. 253118, CS 21900, Evreux Cedex, France) was used to determine the thermal conductivity of the samples (with or without a coating), which were cut into a specific square shape of 40 mm length and 2 mm thickness. The sample was then inserted between two metal blocks, the height of which is adjustable, so as to eliminate the air gap appearing on the sample’s surface (see Figure 10a). The bottom block was set at room temperature, while the upper one was controlled by a heat source in such a way that the temperature difference between these two blocks (∆T) was perfectly regulated at 24 °C. The thermal resistance ($R_{th}$) of the sample can thus be determined in a steady state, i.e., defined as the ratio between ∆T and the induced heat flow (Q) across the sample thickness. Knowing the thickness (e) and the surface (S) of the tested sample, it is possible to infer its thermal conductivity (λ) with the following relationship:(8)$$\lambda =\frac{e}{R_{th}\times S}$$

After 10 min, where the steady state was certainly reached, both values of λ and $R_{th}$ were recorded. All measurements were conducted at room temperature ($T_{amb}=25\;{}^{\circ}C$). To reduce uncertainty, each measurement was carried out on two samples that were assumed to be identical in terms of dimensions and physical properties, as they were cut from the same initial fabric.

Based on the thermal resistance measurements of the neat fabric (denoted as $R_{th\_nf}$) and the coated fabric (denoted as $R_{th\_cf}$), we can determine the thermal conductivity of the coating itself. Simple schemes of the neat and coated samples are illustrated in Figure 10b. Considering that the fabric and the coating layers are thermally connected in serial, $R_{th\_cf}$ is the sum of $R_{th\_f}$ and the resistance of the conductive coating layer ($R_{th\_cond}$):(9)$$R_{th\_cf}=R_{th\_nf}+R_{cond}$$

Eventually, $R_{cond}$ can be deduced based on the measurements of $R_{th\_f}$ and $R_{th\_c}$. As a result, the thermal conductivity of the conductive material (*λ_comp_*) is estimated using the expression in Equation (8).

An alternative approach allows for the prediction of *λ_comp_*, which tends to increase with an increasing particle volume fraction (*v*), and uses the analytical mixing law. The Maxwell model, considered as one of the most commonly used models, has been tailored for composites consisting of a dispersed and a continuous phase. This model was exploited for the thermal conductivity and is given by the following expression, particularly in the case of randomly dispersed particles embedded into a polymer matrix [88,89]:(10)$$\lambda_{comp}=\lambda_{m}\frac{\lambda_{p}+2\lambda_{m}+2v(\lambda_{p}-\lambda_{m})}{\lambda_{p}+2\lambda_{m}-v(\lambda_{p}-\lambda_{m})}$$
where $\lambda_{m}$, $\lambda_{p}$, and $\lambda_{comp}$ denote the thermal conductivity of the matrix, particles, and composite, respectively. The values of $\lambda_{m}$ and $\lambda_{p}$ can be found in Table 5, allowing to deduce $\lambda_{comp}$, as shown in Table 6. Note that the thermal conductivity of the composite inks is close to that of the matrix, as the volume fraction of the particle content is relatively low.

As reported in [90], the specific heat capacity of a composite ($C_{p,comp}$) can be predicted by Equation (6). Actually, $C_{p,comp}$ is modeled as the weighted average of each constituent’s heat capacities in the case of an isotropic material with constant pressure and volume (negligible thermal expansion) with no local strain or stress [91].
(11)$$C_{p,}{}_{comp}=(1-w)C_{p,m}+wC_{p,p}$$
where *w* denotes the weight concentration of the particles dispersed in the inks (Table 1). *C_p_*_,_*_m_* and *C_p_*_,_*_p_* are, respectively, the specific heat capacity of the polymer matrix and the particles, the values of which are found in Table 5. Eventually, $C_{p,comp}$ of the three inks can be estimated (cf. Table 6) thanks to Equation (11). As a result of its higher specific heat, the carbon composite can minimize changes in temperature compared to its copper and silver counterparts. Indeed, it takes a little bit more time for carbon to heat up once the power is on. Inversely, copper and silver cool down faster than carbon.

### 4.3. Electrical Characterization

An electrical characterization of the composite inks was carried out based on the determination of their resistance (denoted as R). This allows for an estimation of the material’s conductivity (denoted as σ) thanks to the following expression:(12)$$\sigma =\frac{l}{R\times S}$$
where S and l, respectively, denote the area and the length of the sample.

There are many methods for determining the resistivity of a material, but the technique may vary depending upon the type of material, the magnitude of the resistance, and the shape and thickness of the material. One of the most common ways of measuring the resistivity of thin, flat materials such as semiconductors or conductive coatings is to use a two-point probe method. Such a technique involves bringing two probes into contact with a material of unknown resistance. A DC current is applied between these two probes, and a voltmeter measures the voltage difference between them. The resistivity is computed from geometric factors, the source current, and the voltage measurement. The Ohm-meter (RM_804) used for taking direct resistance measurements in this work includes a DC current source, a sensitive voltmeter, and a two-point probe (see Figure 11). To enhance the measurement’s precision, several values of resistance are recorded at different distances between the two probes.

The mixing rule, being simple and conveniently used in practice, is an alternative phenomenological approach for an estimation of the electrical conductivity of composite inks $\left(\sigma_{comp}\right)$. The most appropriate form of the mixing rule, which agrees with effective medium models and satisfactorily represents the experimental data, can be written as follows [92]:(13)$$\sigma_{comp}^{\alpha }=(1-v)\sigma_{m}^{\alpha }+v\sigma_{p}^{\alpha }$$
where $\alpha_{m}$ and $\alpha_{p}$ are, respectively, the electrical conductivity of the polymer matrix and the particles, the values of which are available in Table 5. *α* is the power factor that can be determined by fitting Equation (7) to the experimental conductivity values (detailed later in Section 4.3). For an ideal parallel arrangement of particles where the composite exhibits a column-like structure oriented in parallel to the electric field direction, α=1. On the other hand, when the particles are connected in series, where the aligned columns are perpendicular to the field direction, α=−1. As a matter of fact, the *α* value in real composites is determined by the morphology of the composite and falls within the interval −1<α<1. In the case of α→0, the power function described in Equation (7) can be simplified to the linear Lichtenecker formula [93]:(14)$$\log(\sigma_{comp}^{})=(1-v)\log(\sigma_{m}^{})+v\log(\sigma_{p}^{})$$

As pointed out in [94], the electrical conductivity of plastic foams can be represented by the Lichtenecker formula.

### 4.4. Heatmap Characterization

Figure 12 illustrates the experimental setup used to characterize the heating profile of a tablecloth (similar to a car seat, provided by Tesca), which was implemented with the coated fabric developed in this study. The tablecloth is electrically powered by a DC voltage supply of 12 V (Rohde & Schwarz HMP 2020 Amplifier, Rohde & Schwarz GmbH & Co KG, Munich, Germany) through the connection with two principal copper electrodes. The resulting current is then conducted to the carbon patches that naturally produce heat under the Joule effect. This effect is well known as Joule’s law, which states that the power of heating (*P*) generated by an electrical conductor equals the product of its resistance (*R*) and the square of the current (*I*). *I* is measured using a current probe (Fluke 80 Meter Clamp, Fluke Corporation, Everett, WA, USA). For an estimation of the electric–thermal conversion, both the voltage and current are recorded in real time through a DEWE-Soft card SIRIUS. To obtain a more intuitive temperature image of the tablecloth (on the back side as opposed to the coating surface), a thermal infrared camera (Optris Xi400, Optris GmbH, Ferdinand-Buisson, Berlin, Germany) was used during the experiment. Thermal imaging consists of transforming the measurements of the infrared radiation into a radiometric image, reflecting the heatmap characterization of the tablecloth. Actually, each pixel of the radiometric image is converted to a temperature value via complex numerical algorithms. 

## 5. Results and Discussions

### 5.1. Morphological Analysis

Figure 13 illustrates the SEM images of the top view (Figure 13a–c) and the cross-sectional view (Figure 13d–f) for carbon, copper, and silver inks. The focus of this study is to assess the ink quality, including the homogeneity of particle dispersion, and the penetration level of the ink into the fabric substrate (which is related to the fluid viscosity).

As illustrated in Figure 13a, the carbon ink provides a homogeneous printed surface, despite some tiny defects appearing on the coating layer. The copper ink leads to a more rugous surface that possibly results from an inhomogeneous particle dispersion. The clear areas are rich in copper particles, while the dark areas represent the beginning of atmospheric oxidation of the copper ink (Figure 13b). Noted that the particle size (median value) of the copper composite is around 10–20 times larger than that of the carbon composite. This explains why more details are observed for the copper’s particles, especially when the images in Figure 13a,b are shown under the same magnification. As illustrated in Figure 13c, the silver surface, which exhibits some significant defects (in black), is not homogeneous compared to the others. The fact is that the silver ink, with its low viscosity, is not suitable for 3D printing, as it is not viscous enough to be prevented from penetrating the pores of the fabric [95]. As a result, the fabric is not completely covered by the printed silver surface, particularly in the case of a thin coating layer. The presence of significant defects or large holes in the printed fabric demonstrated by SEM analysis have also been mentioned in a previous work [96]. Such a phenomenon might originate from solvent evaporation during the polymerization process as well as from bubbles introduced in the 3D printing process. 

Further details of the morphological properties can be found in the cross-section micrographs. As shown in Figure 13d,e, both the carbon and copper inks reside on the fabric surface, confirming a sufficient binding and adequate viscosity as they appear highly resolved. Several inevitable minor defects such as zigzag burrs and small holds are observed, but on the macroscopic point of view, they should not much affect the electrical properties of the device. In conclusion, the SEM results demonstrate a satisfactory printing quality of the carbon and copper composites, which is essential for the reliability of the heating performance [97]. On the contrary, the silver ink (Figure 13f) exhibits an excessive and irregular spreading to the fabric edges, at which it forms zigzag burrs of the conductive lines. This effect is probably due to the rheological properties of the ink to some extent [98]. In addition, the fabric substrate is a regular multi-scale porous structure and has a rough surface; thus, its surface energy has regular variation. This results in a different propagation of the ink through the inter-thread space with respect to the surface of the fabric [99].

Finally, there is always a gap between the pressurized syringe and the porous substrate during the printing process. On the one hand, the ink spills through the pressurized syringe onto the fabric substrate, resulting in the penetration of the ink in suspension, as well as in the formation of zigzag burrs on the edges of the 3D printed lines. On the other hand, surface roughness is one of the many factors that affect the electrical properties of printed conductor lines. A smooth and uniform surface could allow the formation of continuous conductive lines and prevent short circuits [100,101]. Based on that reason, the copper and carbon inks are considered as the most adequate to construct our final design.

### 5.2. Thermal Analysis

Table 7 shows the measurement values of the thermal resistance ($R_{th}$) and the thermal conductivity (λ) of the neat fabric, as well as the other coated fabrics printed with composite inks made of C/PU, Cu/PU, and Ag/BGA. As observed, the thermal resistance of the coated fabric ($R_{th\_cf}$) is slightly higher than that of the neat sample ($R_{th\_nf}$). Subtracting these values gives the resistance of the conductive coating ($R_{th\_cond})$, according to the model in Equation (9). The coatings are stacked with different numbers of layers, leading to a slight change in $R_{th\_cf}$ (see Table 7). Logically, the higher the layer’s number, the higher the $R_{th\_cf}$ value, which is manifested by an increase in $R_{th\_cond}$ (i.e., proportional to the coating thickness). Table 7 shows the value of $R_{th\_cond}$ calculated with one layer, allowing to determine the thermal conductivity ($\lambda_{comp})$ of the corresponding composite. These values are consistent with those (cf. Table 6) estimated using the mixing law in Equation (10).

The thermal resistance of the coated fabric is revealed to be similar to that of the neat sample, regardless of which composite ink is chosen. Actually, the thickness of the coating layer is much smaller than that of the neat fabric while its thermal conductivity ($\lambda_{nf}$) is in turn smaller than that of the composite ${(\lambda }_{comp})$ (Table 7). Consequently, $R_{th\_cond}$ is negligible with respect to $R_{th\_nf}$, explaining why the coating ink does not affect the thermal conductivity of the fabric (i.e., $\lambda_{cf}\sim \lambda_{nf}$). This assumption is valid under a design condition where the coating is sufficiently thin with respect to that of the neat fabric. 

Table 7 shows the results of measurements performed on samples with different numbers of coating layers (two, three, or four layers), demonstrating that the number of coating layers does not significantly modify the thermal resistance of the fabric. However, it does affect the textile flexibility, which becomes more rigid with more important number of conductive coating layers. As a result, the number of layers was chosen to be smaller than four. On the other hand, as the fabric is quite porous, and a little amount of the ink may penetrate the fabric (see SEM image in Section 5.1), it is preferable to perform more than one coating layer. Based on the best compromise between the mechanical flexibility of the fabric together with the printing quality, the number of coating layers was chosen to be equal to two or three, considered as the most optimized design configuration.

Considering the whole system is powered under the same heat flow (Q) across its thickness, the temperature change (∆T) of the fabric itself is revealed to be much higher than that obtained by the coating. In the other words, the ∆T ratio between the fabric and the coating is equal to the corresponding resistance ratio, as expressed by the following equation:(15)$$\frac{{∆T}_{fabric}}{{∆T}_{coating}}=\frac{R_{th\_fabric}}{R_{th\_coating}}$$

Based on the thermal resistance in Table 7, it is revealed that the above ratio is equal to approximately 20 with the three-layer coating or 60 with the one-layer coating. This means that almost all the heat flux energy is delivered to the fabric, while a very small amount of heat loss is due to the conductive coating. The above result confirms the high efficiency of the developed system for heating a car seat.

### 5.3. Electrical Analysis

Figure 14a illustrates the electrical resistance trends of the three coatings as a function of the specimen’s length, which is defined by the distance between the two-point probes clamped to the samples. Figure 14b displays the conductivity of all coatings, i.e., deduced from the expression in Equation (12).

The copper and particularly the silver composites exhibit extremely small resistance values, reflecting the high electrical conductivity of their corresponding particles, as shown in Table 5. Such small values could provoke high uncertainty measurements, explaining why the resistance is not perfectly linear as a function of the length. As expected in Figure 14a, the carbon composite leads to a substantially higher resistance in comparison to the other two inks, making it efficient in producing the heating Joule effect. For a given moderate current, a higher material resistance gives rise to a higher heating power. Note that the resistance value of the carbon composite is sufficiently higher compared to those of the copper and silver inks, leading to a decrease in the uncertainty measurement. Accordingly, the carbon sample displays a better linear trend in the resistance, making estimation of its electrical conductivity (*σ*) more precise. As seen in Figure 14b, *σ* of the carbon composite is almost constant at different lengths, while those of the copper and silver composites slightly fluctuate. In addition to the uncertainty, this behavior might be somehow caused by the homogeneity of the sample, depending on how particles disperse within the polymer matrix.

Table 8 shows the conductivity of the three composites, calculated as the average of the four values depicted in Figure 14b. This empirical value is then fitted to the model in Equation (8), allowing for a determination of the power factor, α. All samples have similar value of α, which correlates to the results found in the literature. Landau et al. [102,103] derived Equation (8) with α=1/3 for any isotropic two-phase mixture, which is similar to our case. It is worth noting that the mixing rule of Equation (8) with constant *α* agrees with the effective medium models or percolation theory only within a narrow concentration range. In other words, *α* must be different if being outside of this range. For instance, at v→0 (composite with no particle content) and v→1 (composite with 100% particle content) for spherical particles, α=2/3 and α=−1/3, respectively. Accordingly, the theoretical value α=1/3 is the arithmetic mean of the two limiting values [102,103]. As the composite inks used in this study were made with a low particle concentration (less than 20%), the factor *α* is supposed to be constant. Finally, these results may be helpful for a deeper understanding of the combined structure of fillers in polymer matrices, which will expand the scope of applications for these materials.

Accordingly, the electrical analysis allows us to conclude that both copper and silver composites exhibit interesting properties that can be used as electrical electrodes with good electrical conductivity. The carbon composite, however, is more suitable to be employed as a heating resistor, thanks to its reasonable resistance value (i.e., not so high to be able to conduct electricity, and not so low to dissipate enough Joule heating). The thermal analysis reveals that all the composites have the ability to favor heat conductivity, while not drastically altering the thermal properties of the fabric. Finally, the morphological analysis via SEM shows the poor quality printing of the silver ink, justifying its absence in the design of a real prototype.

To evaluate the dispersion of conductive inks, measurements of the electrical resistance were carried out on seventeen identical samples printed with carbon or copper coatings. The boxplot in Figure 15 provides a statistical overview of the results acquired from these samples, comprising locality, dispersion, and skewness of the acquired data. The numerical values, consisting of the mean value, variation coefficient (denoted VC, which is equal to the ratio between the standard deviation (SD) and the mean), and maximum and minimum values, are summarized in Table 9. The three quartiles are reported as well, where $Q_{1}$ is the 25th percentile (also called the lower quartile), $Q_{2}$ is the 50th percentile (i.e., the median of the entire dataset), $Q_{3}$ is the 75th percentile (also called the upper quartile), and IQR is the interquartile range. To better assess the variability of the data, we provide here an estimation of the quartile coefficient of dispersion (QCD), given by Equation (16). The higher the QCD value, the more dispersed the dataset.
(16)$$QCD=\frac{Q_{3}-Q_{1}}{Q_{3}+Q_{1}}=\frac{IQR}{Q_{3}+Q_{1}}$$

Similar to the above discussion, the copper coating leads to highly dispersed data, where the electrical conductivity (σ) lies between 0.5 and $2\times 10^{4\;}S.m^{-1}$. On the other hand, the carbon coating exhibits a smaller dispersion, with the range of $3.3–4.6\times 10^{2\;}S.m^{-1}$.

The detailed data related to the box graph given in Table 9 lead to the following conclusions:Regarding the mean and median values of the electrical conductivity, it is clear that the copper composite has a higher electrical performance than the carbon composite.In all cases, the data distributions are asymmetrical, as the median (horizonal lines inside the whisker box) was revealed to be higher than the mean (the cross). As a result, the distributions obtained from both copper and carbon samples were skewed to the left (or negative skew), appearing as right-leaning curves.No measurements exhibited any outliers or extremes values (i.e., falling below $Q_{1}-1.5\;IQR$ or above *Q*_3_ + 1.5 *IQR*), meaning that the highest and lowest occurring value were within this interval.A finer analysis regarding the data dispersion via two relative coefficients, VC and QCD:The VC and QCD coefficients were computed in different manners but had similar values, regardless of which composite was studied. Both coefficients are considered as relevant indicators that allow for an efficient analysis of the data variability.The electrical measurements of the copper coating exhibit significantly higher dispersion than those of the carbon coating, which is ultimately related to the measure precision.In the carbon coating, both QCD and VC values are relatively low, confirming a good repeatability of the data (<7%). This is contrary to the case of the copper coating, where these coefficients were found to be higher than 20%. This is as a result of the small resistance values of the copper, which in turn drastically increase the measurement uncertainties.

### 5.4. Heatmap Analysis

In this study, the optimized design coating (D2) is selected to be printed on a fabric and implemented into a real tablecloth. A heatmap analysis on the back side of the tablecloth was performed using the experimental setup described in Section 4.4. The real prototype was made of conductive coatings comprising 3-layer carbon patches and 2-layer copper electrodes, similar to the model built in the FEM. Thanks to the optimized design and adequate materials of the smart coating, the heating tablecloth successfully matches the specifications defined by Tesca.

As graphically illustrated in Figure 16a, the temperature of the tablecloth surface was almost homogeneous for all three areas, without the presence of significant hot spots. This finding is in agreement with the results obtained by the numerical simulation shown in Figure 16b. For a better comparison between the FEM and the empirical data, Table 10 summarizes the average temperature value together with the deviation coefficient (DT3) for the three areas. Remarkably, the empirical measurement demonstrates the same temperature for all these areas, with very small DT3s of 0.2–0.5%. For both areas 1 and 2, this deviation is somewhat higher in the FEM, i.e., approximately 0.7–1.2%. In either experiment or simulation, area 1 gives a higher value of DT3 than area 2. This is obvious as the temperature of area 2 is closer to that of area 3, which is in turn nearest to the electrical source. Consequently, the temperature gradually decreases from area 3 to 1.

The discrepancy between experiment and simulation can be assessed through another relative deviation coefficient (denoted $D_{e-s}$), given by:(17)$$D_{e-s}=\frac{T_{e}-T_{s}}{T_{e}}\times 100\%$$
where $T_{e}$ and $T_{s}$ are the temperatures of a given area obtained from experiment and simulation, correspondingly. For all the three areas, $D_{e-s}$ is relatively small, reflecting a good correlation between the numerical model and the practical data. Similar to the trend of the DT3 coefficient, $D_{e-s}$ decreases (from 2.5% down to 1.6%) from area 3 to area 1. 

In the following, the heating behavior of the coated fabric was investigated under different voltage levels (denoted U), from 3 V to 18 V. Current measurements were also performed (denoted I), which allows determination of the electrical power (i.e., P=U×I). 

Figure 17a illustrates the time evolution of the temperature, based on which the response time can be determined. All raw and estimated data obtained from experiments are summarized in Table 11. Under 12 V, the temperature reaches 43 °C after a response time of 112 s. With higher applied voltages (e.g., 15 V and 18 V), the temperature increases but the response time is unchanged. Logically, a lower voltage leads to a faster heating process, but the temperature is not sufficient enough for the user’s comfort, regarding the criteria imposed by our automobile partner. 

As expected, in Figure 17b, the electrical power (denoted P) is the square of the input voltage (U²). Based on the Joule effect of a conductor, the ratio $U^{2}/P$ gives an estimation of the electrical resistance of the two copper power buses, which is revealed to be equal to approximately 5.5 Ω. This is rational, as each copper electrode was designed with a resistance value of 2–3 Ω. Figure 17b also confirms a square behavior between the temperature change and the applied voltage. Actually, the temperature change is supposed to be linear to the heat flow (Q), which is ultimately related to the electrical power due to the Joule heating effect. 

Based on the specifications defined by Tesca, the applied voltage was chosen as 12 V, leading to a power electric consumption (${Փ}_{J}$) of approximately 25.5 W (cf. Table 11). As the thermal resistance ($R_{th\_tab}$) of the tablecloth was equal to 1.34 $W^{-1}.K$ (found in Section 3.3), the rate of conductive heat transfer ($P_{cond},inW$) to achieve the limit temperature ($T_{l}\to 43\;{}^{\circ}C$) is equal to:(18)$$P_{cond}=\frac{T_{l}-T_{f}}{R_{th\_tab}}≃14.2\;W$$

On the other hand, the rate of convective heat transfer ($P_{conv},\;in\;W$) is computed as:(19)$$P_{conv}=h_{c}S\;(T_{l}-T_{0})≃4.8\;W$$

As a result, $P_{cond}+P_{conv}\sim 19\;W$. The efficiency of the heated tablecloth is thus estimated to be equal to approximately 75%. Energy losses (~25%, i.e., not considered in Equation (3)) probably stem from the Joule heating dissipated by the two principal copper electrodes and thermal leakage through the materials caused by a connection between the fabric and the coating layers. In reality, a car seat is in contact with a human body, thus no more convection phenomena occur. Thermal conduction between the heated seat and the body is employed instead. Such an issue should be considered in the future steps of this work.

### 5.5. Preliminary Results of Aging Performances

In order to verify the durability and stability of the coatings with respect to the environmental changes, three aging tests were carried out including: (1) a mechanical test, (2) a heating test, and (3) a moisture test.

The mechanical test consisted of applying a load of 890 N (i.e., corresponding to the typical weight of an adult male) for 200,000 cycles at rate of 100 cycle/min. The test was carried out based on the European SAE Standard using J826 H-point manikin [103] (ETD technical trading, Seeveta, Germany). Temperature measurements were performed on the three areas of the carbon surfaces via a thermal camera for both sewn and non-sewn samples (Figure 18). The experimental results revealed that after the mechanical solicitations, all areas of the non-sewn fabric achieved the desired temperature of around 43 ± 0.5 °C. This confirms the good mechanical behavior of the printed coating under such important cycles of the aging test. Interestingly, this test did not much affect the thermal behavior of the coated fabric sewn at the center. However, sewing could lead to a somewhat increased temperature (~50 °C) because of the higher electrical current in the restricted section of the conductive coating. Design and process improvements will be further explored to determine the cause of such a critical issue. 

Figure 19a illustrates the one-cycle profile (i.e., equivalent to a period of 12 h) of the temperature and moisture content applied to the conductive coatings made of a copper or carbon composite. The temperature tests were carried out for a short duration of 24 h or 48 h. The moisture test, however, lasted much longer (12 days, i.e., comprising 24 cycles). 

As seen, the thermal cycle is composed of five steps as described below. For each step, the temperature is

Increased from 20 °C to 80 °C for 1 h at a rate of 1 °C/min;Kept stable at 80 °C for 4 h;Lowered to −40 °C at a rate of 1 °C/min for 2 h;Maintained constant at −40 °C for 4 h;Increased up to 20 °C at a rate of 1 °C/min for 1 h.

Additionally, the moisture cycle consists of four steps as described below. For each step, the moisture content is

Increased from 30% to 80% at a rate of 0.83%/min for 1 h;Maintained at 80% for 4 h;Decreased to 30% at a rate of 0.83%/min for 1 h;Maintained at 30% for 6 h.

The electrical conductivity (σ) of the samples was measured before and after the thermal and moisture tests, to better highlight their impact on the printed coating performance. Figure 19b displays the variation in σ in the case of the copper samples. Actually, a very small change (∆σ<1%) was observed for the carbon coatings, which is thus assumed to be negligible in this analysis. For an easier comparison, the electrical conductivity shown in Figure 19b was normalized with respect to its value found before the tests. It has been pointed out that both thermal and moisture tests have an impact on the electrical properties of the copper coating. σ clearly decreased by around 30% after 12 days of moisture treatment, while it decreased by approximately 20% and 40% after 24 h and 48 h of the thermal procedure, respectively. This result of the aging tests clearly demonstrated that the copper composite used in this study was not very stable with respect to environmental changes, in particular in extreme conditions such as excessive negative and/or positive temperatures together with high moisture content. Finding an appropriate polymer matrix to protect the conductive coating would be an effective solution to enhance its aging performance. This matter is of primary importance and thus will be further investigated in our future research. 

## 6. Conclusions

In this study, we proposed a new generation of car seat heaters based on an innovative flexible electronic design combined with 3D printing additive manufacturing (AM). The fabric substrate was coated with different layers of conductive materials with the intention of providing a regular temperature up to 41–43 °C in a fast response time without overheating. Morphological characterizations based on SEM observations confirmed the printing quality and verified whether or not the selected composite inks were applicable to the AM method. Electrical and thermal characterizations were conducted through experiments and analytical mixing rules, with the aim of identifying the pertinent parameters of the materials. These findings were not only needed for the FEM implementation via COMSOL software, but also allowed for a better comprehension of the design strategy. The copper composite ink, thanks to its excellent electrical conductivity, was chosen to be the electrode material, while the carbon composite was more suitable to be employed as a heating resistor. The silver ink, because of its high cost and unsuitable viscosity, was not further considered in the practical prototype. A heatmap characterization was performed using COMSOL simulations, which was then conducted in a real tablecloth setup. A good agreement between numerical results and empirical measurements was observed. Optimization of the critical connections between the copper and carbon coatings led to obtaining the perfect homogenous temperature over the whole fabric surface and definitively solved the hot spot issue. 

After the successful development and integration of the heating device in its expected configuration, further studies were undertaken, principally focusing on aging analyses of the coatings under different external conditions, including temperature, humidity, and mechanical solicitation (dynamic and static regimes). It is believed that optimizing the design structure and material properties, together with an enhancement in the printing quality, will definitively promote the development of heated textiles toward future applications.

## Figures and Tables

**Figure 1 micromachines-14-00762-f001:**
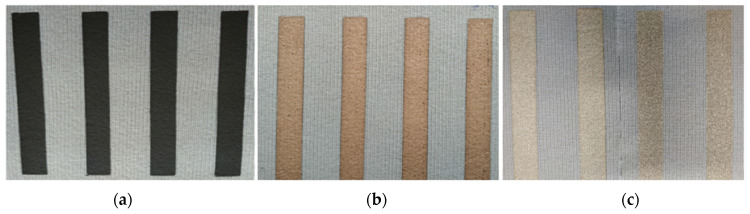
3D printing of different conductive inks made of (**a**) carbon, (**b**) copper, and (**c**) silver coated on the textile substrate.

**Figure 2 micromachines-14-00762-f002:**
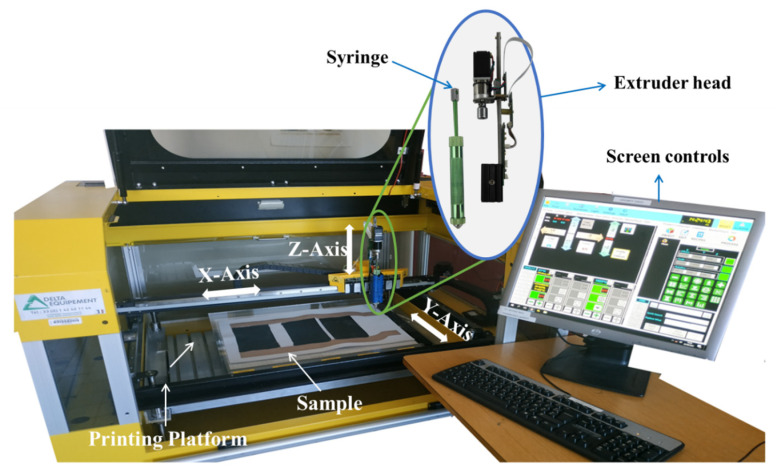
Cross-sectional illustration of the 3D printing process.

**Figure 3 micromachines-14-00762-f003:**
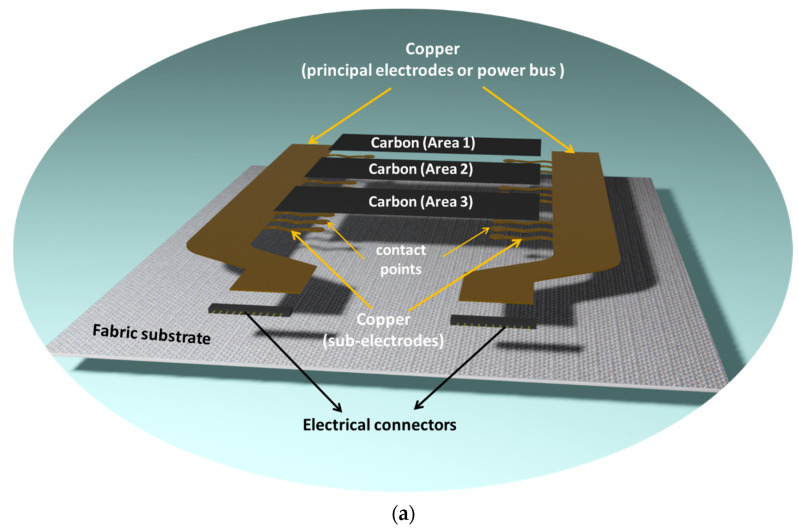
Design patterns of the heating fabric: (**a**) Heating system composed of copper tracks and carbon patches built on a fabric substrate. (**b**) 2D dimensions of the heater design. (**c**) Equivalent electrical scheme.

**Figure 4 micromachines-14-00762-f004:**
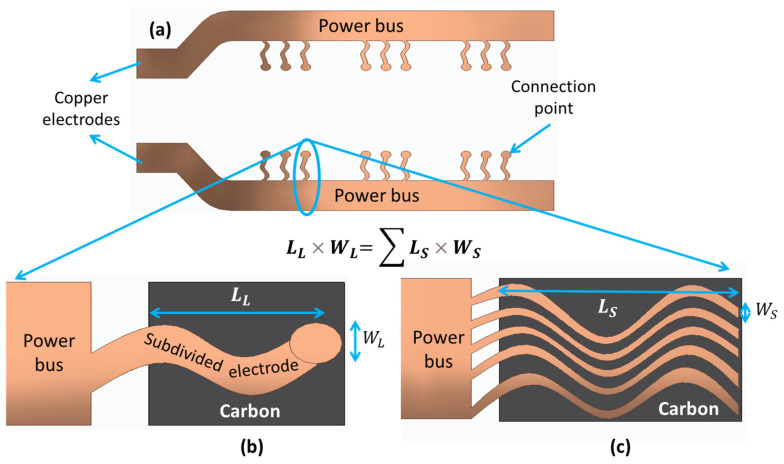
Optimization of the printed coatings: (**a**) initial design; (**b**) zoomed-in view of the connection between the copper electrode and the carbon patch; and (**c**) optimizations performed on designs 1 and 2.

**Figure 5 micromachines-14-00762-f005:**
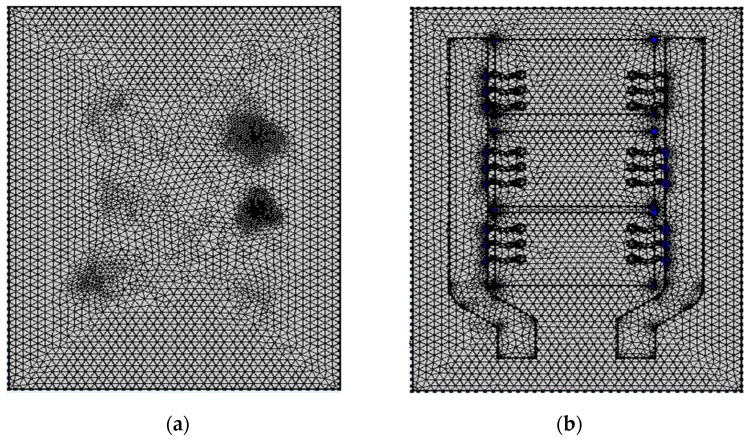
Mesh built in COMSOL Multiphysics for the model geometry of the tablecloth: (**a**) face of fabric substrate (back side) and (**b**) coating face (front side).

**Figure 6 micromachines-14-00762-f006:**
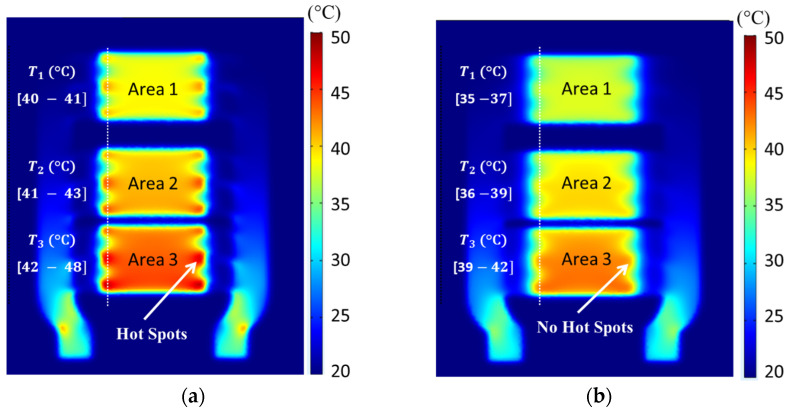
Influence of the thermal conductivity (λ) on the numerical heatmap of the tablecloth set: (**a**) $\lambda_{carbon}$ and $\lambda_{copper}$ values shown in Table 3; and (**b**) $\lambda_{carbon}=\lambda_{copper}=10\;W{.m}^{-1}.K^{-1}$.

**Figure 7 micromachines-14-00762-f007:**
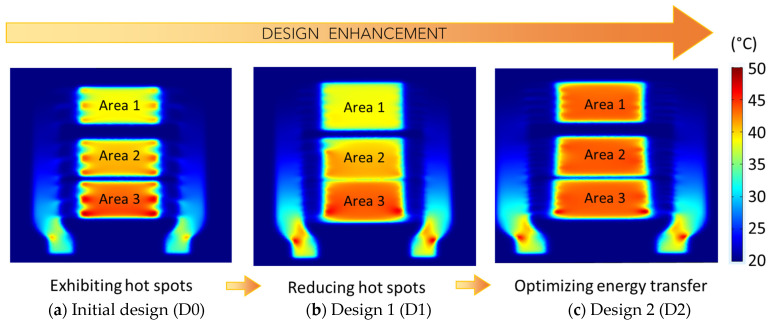
Design enhancement for reducing hot spots and obtaining a uniform heated surface.

**Figure 8 micromachines-14-00762-f008:**
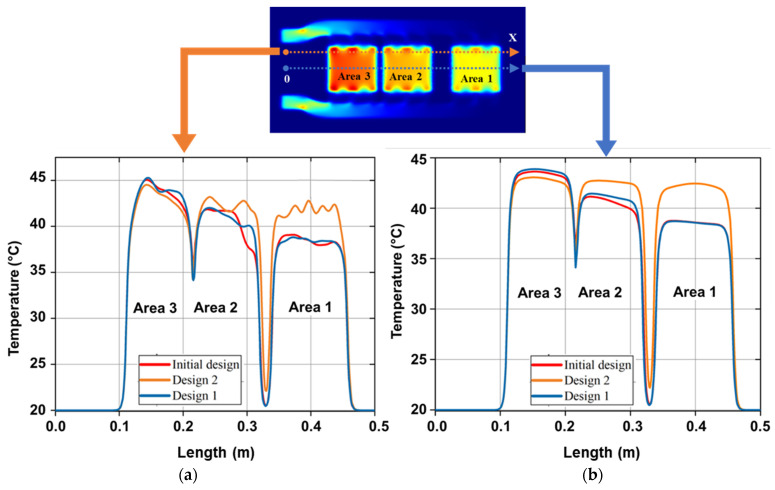
Temperature evolution along the x-axis: (**a**) lateral temperature and (**b**) central temperature.

**Figure 9 micromachines-14-00762-f009:**
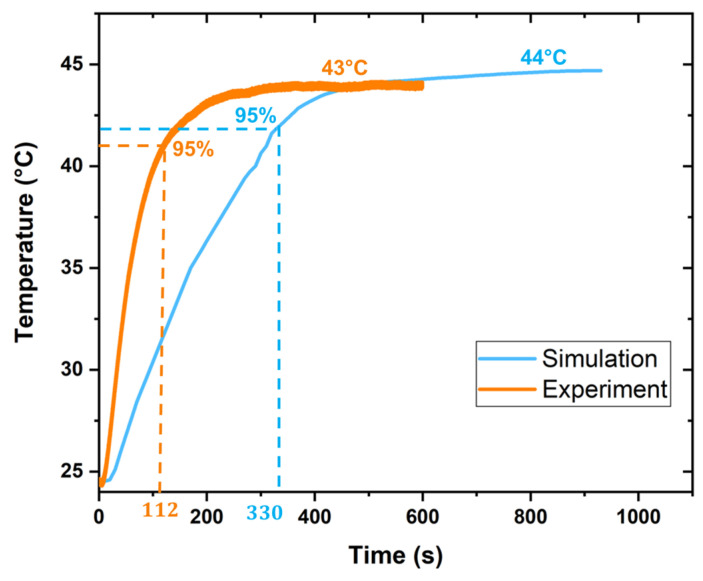
Time evolution of the measured and simulated temperatures performed on the tablecloth with an optimized design.

**Figure 10 micromachines-14-00762-f010:**
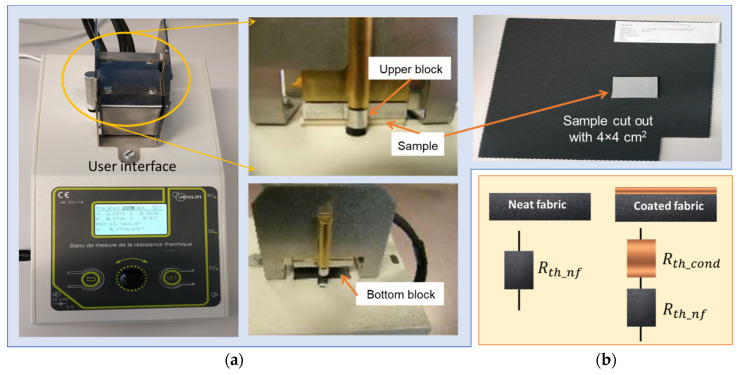
Thermal characterization of fabrics with and without printed coatings: (**a**) Thermal resistance measurement using a Jeulin device and (**b**) equivalent scheme of a simple thermal model.

**Figure 11 micromachines-14-00762-f011:**
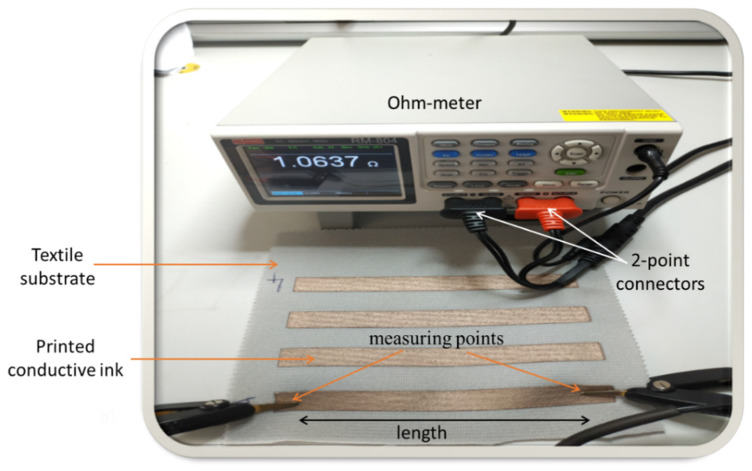
Experimental setup used for measurement of the electrical resistance.

**Figure 12 micromachines-14-00762-f012:**
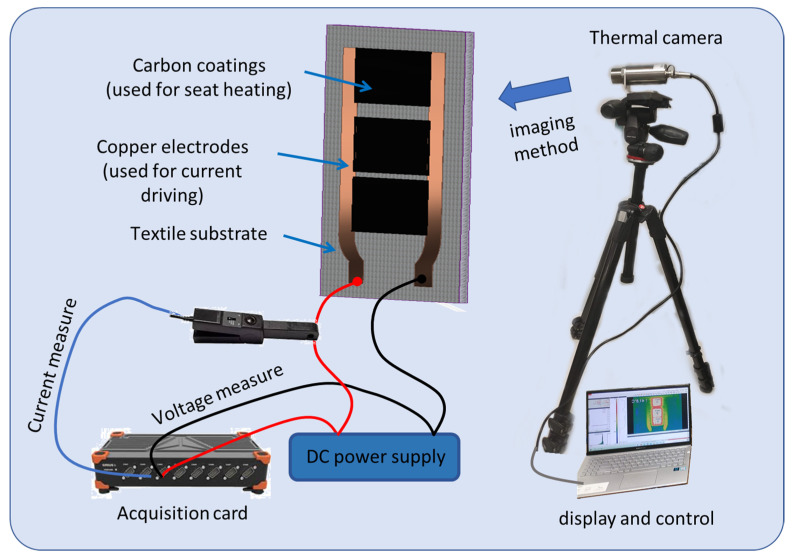
Schematic illustration of the measuring bench used for the thermal characterization of a heated tablecloth.

**Figure 13 micromachines-14-00762-f013:**
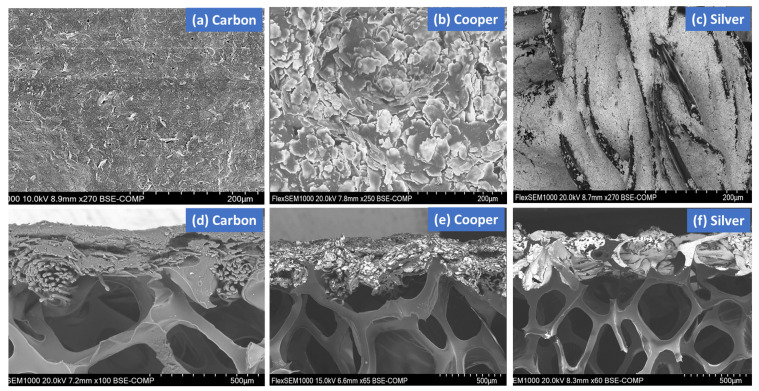
Micrographs of carbon, copper, and silver coatings for (**a**–**c**) the top view (taken on the top surface) and (**d**–**f**) the edge view (taken in the cross-section).

**Figure 14 micromachines-14-00762-f014:**
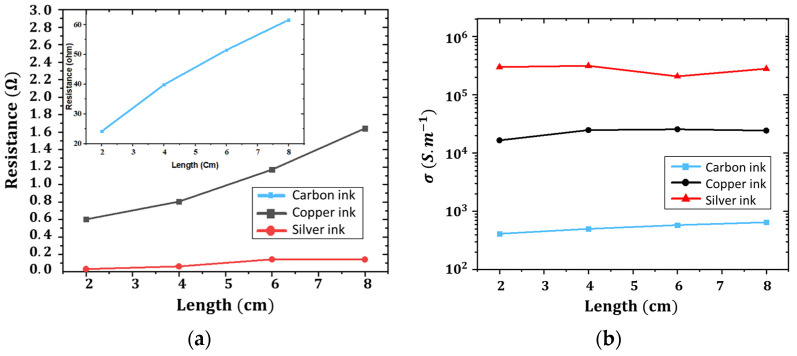
Evolution of the electrical property as a function of the length of different conductive inks: (**a**) electrical resistance and (**b**) electrical conductivity.

**Figure 15 micromachines-14-00762-f015:**
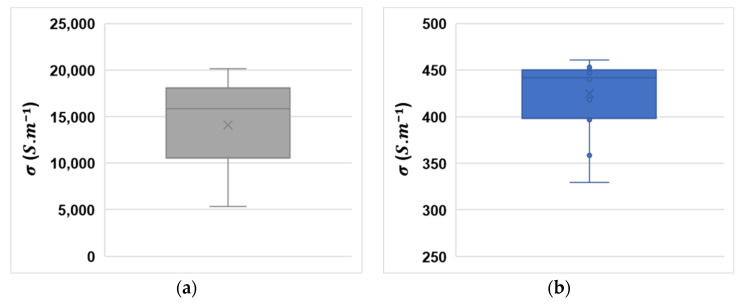
Whisker-box graphs computed from 17 samples coated with (**a**) copper or (**b**) carbon composites.

**Figure 16 micromachines-14-00762-f016:**
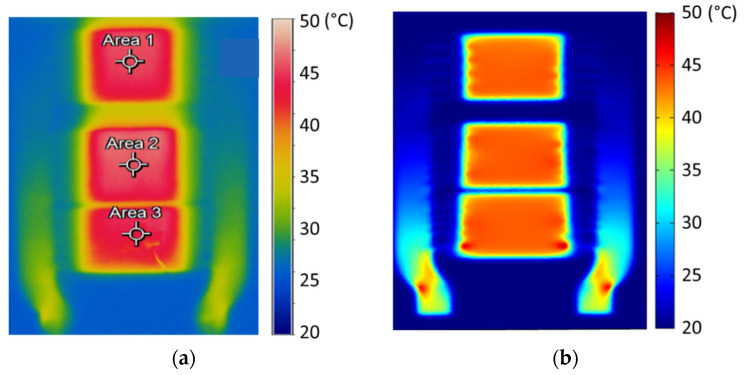
Heat map of (**a**) a real 3D-printed prototype using a thermal camera and (**b**) numerical model using COMSOL software.

**Figure 17 micromachines-14-00762-f017:**
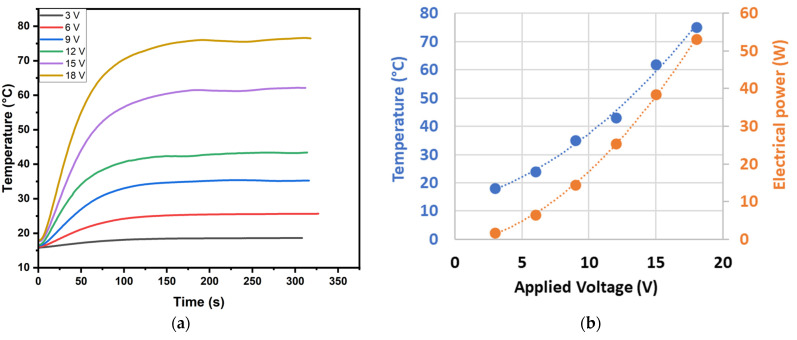
Thermal behavior under different applied input voltages: (**a**) time evolution of temperature behavior and (**b**) temperature (blue curve) and electrical power (orange curve) versus input voltage. Dotted lines represent the quadratic models.

**Figure 18 micromachines-14-00762-f018:**
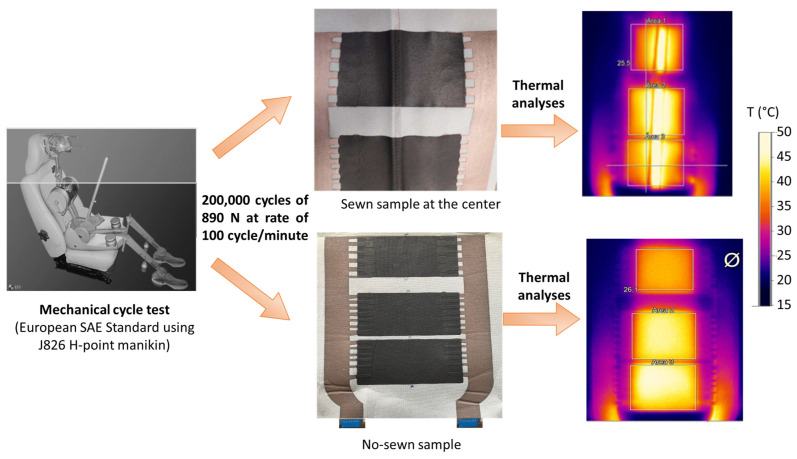
Thermal analyses of both sewn and non-sewn coated fabrics after several mechanical-cycle aging tests.

**Figure 19 micromachines-14-00762-f019:**
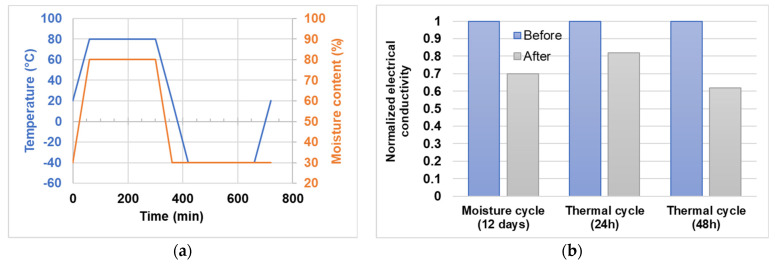
Aging test of the copper electrode coating: (**a**) profile of temperature and moisture in one cycle and (**b**) evolution of the electrical conductivity (normalized value) before and after the temperature and moisture stability tests.

**Table 1 micromachines-14-00762-t001:** General properties of commercially available conductive inks.

Properties	C/PU Ink	Cu/PU Ink	Ag/BGA Ink
Polymer matrix	Polyurethane (PU)	Polyurethane (PU)	Butyl Glycol Acetate (BGA)
Particles	Carbon (C)	Copper (Cu)	Silver (Ag)
Particle size	≤2.5 µm	30–40 µm	≤10 µm
Viscosity (Pa.s)	6.0–7.0	6.5–7.0	0.15
Curing time at 120 °C (min)	2	3	15
Particle content (wt.%)	30	40	62
$\mathrm{Mass}\;\mathrm{density}\;(g/{cm}^{3})$	1.33	1.74	2.16
Particle content (vol.%)	17.6	7.8	12.9

**Table 2 micromachines-14-00762-t002:** Tuning parameters of a 3D printing process [63].

Parameter	Value
Layer Height (mm)	0.2
Number of Skirt Loops	0
Printing Speed (mm/s)	15
Travel Speed (mm/s)	100
Extrusion Multiplier	1
Nozzle Diameter (mm)	0.4
Retraction (mm)	2
Advanced	1.5

**Table 3 micromachines-14-00762-t003:** Measured thermal conductivity of coated and neat fabric samples.

Materials	Dimension W×L×H (mm^3^)	Thermal Conductivity λ $\left(W{.m}^{-1}.K^{-1}\right)$	Heat Capacity $C_{p}\;(J{.kg}^{-1}.K^{-1})$	Electrical Conductivity $\sigma \;(S{.m}^{-1})$	Density $\rho \;({g.cm}^{-3})$	Surface Emissivity ε
Fabric	370 × 500 × 2	0.062	1200	1.0 × 10^−9^	0.8	0.9
Copper	40 × 440 × 0.6	0.367	1052	1.43 × 10^4^	1.74	0.5
Carbon	100 × 170 × 0.4	0.296	1266	4.23 × 10^2^	1.33	0.9

**Table 4 micromachines-14-00762-t004:** Comparison of the heating performance of the three designs.

Area	$T_{mean}\;({}^{\circ}C)$			DT3 (%)			RV (%)		
	D0	D1	D2	D0	D1	D2	D0	D1	D2
**1**	38.8	38.6	42.5	11.2	12.2	1.2	6.5	3.4	1.9
**2**	40.8	41.4	42.8	6.5	5.9	0.7	6.3	5.1	1.8
**3**	43.6	43.0	43.1	0	0	0	9.0	2.9	2.2

**Table 6 micromachines-14-00762-t006:** Thermal properties of the composite inks based on mixing laws.

Ink Composites	*λ* $\left(W{.m}^{-1}.K^{-1}\right)$	*C_p_* $\left(J{.kg}^{-1}.K^{-1}\right)$
C/PU composite	0.377	1266
Cu/PU composite	0.289	1052
Ag/BGA composite	0.274	908

**Table 7 micromachines-14-00762-t007:** Measured thermal conductivity of coated and neat fabric samples. The resistance and the conductivity of the conductive coatings were deduced from Equations (8) and (9).

	*R_th_* $\left(W^{-1}.K\right)$	*λ* $\left(W{.m}^{-1}.K^{-1}\right)$
Fabric sample without conductive coating	20.16	0.062
Fabric samples with conductive coating of:		
3-layer C/PU	21.18	0.065
4-layer C/PU	21.55	0.064
2-layer Cu/PU	21.02	0.066
3-layer Cu/PU	21.45	0.064
3-layer Ag/BGA	21.52	0.064
Conductive inks (1 layer)		
C/PU	0.34	0.367
Cu/PU	0.42	0.296
Ag/BGA	0.45	0.280

**Table 8 micromachines-14-00762-t008:** Measured electrical conductivity (*σ*) and fitting factor (*α*).

Ink Composites	*σ* $\left(S{.m}^{-1}\right)$	*α* (Fitting Factor)
C/PU composite (17.6 vol%)	4.23 × 102	0.317
Cu/PU composite (7.8 vol%)	1.43 × 104	0.307
Ag/BGA composite (12.8 vol%)	1.88 × 105	0.323

**Table 9 micromachines-14-00762-t009:** Statistical data related to electrical properties of the conductive coatings. All parameters have the same unit of electrical conductivity $\sigma \;(S.m^{-1})$, except the two relative coefficients VC and QCD, which are in %.

$$\sigma \;(S.m^{-1})$$	Mean	VC	Min	Max	*Q* _1_	*Q* _2_	*Q* _3_	IQR	QCD
**C/PU composite**	4.3 × 10^2^	28.7%	3.3 × 10^2^	4.6 × 10^2^	4.0 × 10^2^	4.4 × 10^2^	4.5 × 10^2^	5	23.7%
**Cu/PU composite**	1.4 × 10^4^	6.9%	5 × 10^2^	2.0 × 10^4^	1.1 × 10^4^	1.6 × 10^4^	1.8 × 10^4^	7 × 10^2^	5.9%

**Table 10 micromachines-14-00762-t010:** Comparison between measured and theoretical temperatures of three heating areas.

Area	$T_{mean}\;({}^{\circ}C)$		DT3 (%)		$D_{e-s}\;(\%)$
	Experiment	Simulation	Experiment	Simulation	
1	43.6	42.5	0.5	1.2	2.5
2	43.7	42.8	0.2	0.7	2.1
3	43.8	43.1	0	0	1.6

**Table 11 micromachines-14-00762-t011:** Electrical power and temperature in the steady state were obtained with different input voltage levels. The response times during which the system reaches 95% of the steady temperature are also provided.

Voltage (V)	Current (A)	Power (W)	Temperature (°C)	Response Time 95% (s)
3	0.7	1.7	18	65
6	1.1	6.6	24	70
9	1.62	14.5	35	100
12	2.13	25.5	43	112
15	2.56	38.5	62	115
18	2.95	53.1	75	115

## Data Availability

Not applicable.

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
