# Peer review of "Development and Optimization of 3D-Printed Flexible Electronic Coatings: A New Generation of Smart Heating Fabrics for Automobile Applications"

_micromachines, 2023, doi:10.3390/mi14040762_

Round 1
Reviewer 1 Report
In this paper, author have tried to develop smart heating fabric for automotive seat covers. However, for making it suitable for publication, more detail study is needed. Authors must incorporate the below suggestions in the revised draft for considering for further evaluation.
i) Authors stated that “Each conductive line consists of one or several printed thin layers, i.e., approximately 200 μm/layer”. Heating performance will strongly depend on the number of layers. Why it was varied? Author should conduct this study using a particular thickness and fixed number of passes. Before that they should optimize the thickness. They should also mention the coating addon or solid addon after coating.
ii) From the SEM images it can been seen that bonding of coating ink is not very good with the fabric, which has created a doubt on the durability. Author should do further study and report the durability of coating and the heating performance of the coated fabric with respect to flexing, rubbing/abrasion, strain and washing.
iii) Though the coating of conductive ink improved the electrical conductivity, there was no improvement in thermal conductivity. How this thing can be useful in heating fabric?
iv) More detail characterization is required to explain the Joule heating effect discussing the effect of varying applied voltage, power density and time on the change of temperature of fabric.
v) Authors should read the whole manuscript to eliminate the mistakes.
For example:
Page 3, Line 123: there is line written as “Moreover, they are adaptable to various types of flexible substrates such as polyamide, polyester, polyethylene terephthalate (PET), photo- graphic paper among others.” – polyester and polyethylene terephthalate (PET) both are same - not different material.
vi) Why there is almost 25% self citation? It should be reduced.
Reviewer 2 Report
The authors reported a heating technique for car-seat fabric based on the use of smart conductive coatings. An extrusion 3D printer is employed to achieve multilayered thin films coated on the surface of the fabric substrate. The result is good and interesting, and it can be accepted after minor revision. The following is the specific suggestions to improve the manuscript.
1. How to evaluate the dispersion ability of conductive ink? Please give more characterizations.
2. The sheet resistance should be measured.
3. The joule heating performance of the fabric should be measured, especially under various applied voltage (1V, 2V, etc.).
4. The joule heating performance of the conductive fabric should be compared with other E-textiles, such as 10.1002/smll.202208134, 10.1016/j.jcis.2021.06.043, 10.1039/D0NR07433K,10.1016/j.compositesa.2021.106700,10.1016/j.jallcom.2022.167964.
Reviewer 3 Report
In this work, the authors proposed a car seat heater based on flexible electronic design, which is manufactured by 3D printing technology. The manuscript is well organized and systematically summarized. In my opinion, the manuscript can be accepted after following minor revisions.
1. In this paper, copper composite ink is used as electrode and carbon composite material as heating resistor, What is the heat treatment method of copper composite ink and carbon composite material ? How to solve the oxidation reaction of copper wire electrode?
2. It is recommended to add a test for adhesion between the heating electrode and the fabric.
3. Car seat heaters are used frequently, so it is recommended to add heating cycle stability test and environmental stability test
4. It is suggested that the author should quote the following highly relevant literatures. Advanced Materials, 2019, 31(32): 1902479; Advanced Materials, 2021, 33(21): 2007772; Advanced Science, 2022, 9(14): 2105331; Small, 2022, 18(17): 2107811.
